# Prevention and Management with Pro-, Pre and Synbiotics in Children with Asthma and Allergic Rhinitis: A Narrative Review

**DOI:** 10.3390/nu13030934

**Published:** 2021-03-14

**Authors:** Lien Meirlaen, Elvira Ingrid Levy, Yvan Vandenplas

**Affiliations:** KidZ Health Castle, UZ Brussel, Vrije Universiteit Brussel, 1090 Brussels, Belgium; lienmeirlaen@hotmail.com (L.M.); elvira.levy9@gmail.com (E.I.L.)

**Keywords:** probiotics, prebiotics, synbiotics, microbiome, children, allergic rhinitis, asthma

## Abstract

Allergic diseases including allergic rhinitis and asthma are increasing in the developing world, related to a westernizing lifestyle, while the prevalence is stable and decreasing in the industrialized world. This paper aims to answer the question if prevention and/or treatment of allergic rhinitis and asthma can be achieved by administrating pro-, pre- and/or synbiotics that might contribute to stabilizing the disturbed microbiome that influences the immune system through the gut–lung axis. We searched for relevant English articles in PubMed and Google Scholar. Articles interesting for the topic were selected using subject heading and key words. Interesting references in included articles were also considered. While there is substantial evidence from animal studies in well controlled conditions that selected probiotic strains may offer benefits in the prevention of wheezing and asthma, outcomes from clinical studies in infants (including as well pre- and postnatal administration) are disappointing. The latter may be related to the multiple confounding factors such as environment, strain selection and dosage, moment of administration and genetic background. There is little evidence to recommend administration of pro, pre- or synbiotics in the prevention of asthma and allergic rhinitis in children.

## 1. Introduction

### 1.1. Prevalence of Asthma and Allergic Rhinitis

The global prevalence of atopic diseases such as asthma, allergic rhinitis and atopic dermatitis is remarkable and has been expanding over the years [1]. Allergic rhinitis occurs in 10 to 30% of adults and up to 40% in children and its prevalence is increasing [2]. With around 339 million people affected globally, asthma is one of the most common long-term non-transmissible diseases [3]. The worldwide prevalence of doctor-diagnosed asthma in adults is 4.3% (95% confidence interval (CI) 4.2–4.4), with a wide variation between countries: the highest occurrence is found in developed countries such as Australia (21%) and the lowest in third world countries such as Ethiopia (2%) [4]. In children, asthma is more frequent in boys than in girls due to their smaller airways relative to their lung size, with a turnaround during puberty, as the prevalence in women is 20% higher than in men [5]. Asthma prevalence is steady or even shrinking in many developed countries, but as lifestyles become more westernized in developing countries, there is a fast increase in its prevalence in these parts of the world [6]. The interaction between the genomic background, changing environmental conditions such as more pollution [7], increasing obesity, the “hygiene hypothesis” and less breastfeeding [8] is likely to play a crucial part. Parental reduction in smoking has proven to reduce asthma [9]. Important to mention is that in less developed countries, the detection rate of allergic disease is likely to be lower, which may result in an underestimation of its prevalence [10]. By identifying and characterizing more of these conditions and the involved lifestyle factors, epidemiologic studies try to deduce potential approaches for prevention of allergic diseases [11]. Asthma causes impaired life quality, substantial disability and preventable deaths in children and adolescents, combined with important health care costs [6]. As a consequence, the increased social and economic burden of asthma makes asthma prevention an important public health goal [12].

### 1.2. Pathophysiology Asthma and Allergic rhinitis

Atopic diseases like asthma and allergic rhinitis are complex multifactorial conditions of which the outcome is strongly influenced by a complex interplay between genetic background, the state of the body’s defenses, gut microbiota and the environment. There are different mechanisms and typical pathological characteristics of asthma immunopathology, which can be divided in three groups: non-eosinophilic (neutrophilic type 1 and type 17 and pauci-granulocytic), eosinophilic (allergic and non-allergic), and mixed granulocytic inflammation [6]. The eosinophilic group represents 50% of all asthma patients. In this process, allergen or trigger factor exposure stimulates local inflammatory responses mediated by immunoglobulin E (IgE) release. This leads to allergen sensitization and the forming of an atopic response. Type 2 T helper (Th2) cells play a crucial part in this inflammatory process by producing cytokines that control fabrication of allergen-specific immunoglobulin E and inflammation of tissue characterized by the invasion of eosinophils, mast cells and activated CD4+ T-cells. Regulatory T-cells (Treg) are involved in preventing the sensitization to allergens by the production of anti-inflammatory cytokines such as IL-10, by secreting transforming growth factor B, and by possibly suppressing the production of immunoglobulin E and proliferation of Type 1 T helper (Th1)/Type 2 T helper (Th2) balance. The mechanisms of tolerance induction are complex [13]. The intestinal microbiome contributes to the pathological process of allergic diseases because of its notable effect on mucosal immunity. A healthy microbiome at a young age changes the balance between T helper 1 T helper 2, shifting towards a T helper 1 cell response. About 60–70% of the immune cells are located within the gastrointestinal tract. On the other hand, atopic diseases involve Type 2 T helper reactions to allergens. Unusual allergic responses are believed to occur in cases of intestinal dysbiosis during the development of the immune system, causing a shift of the Th1/Th2 cytokine balance towards a Th2 response, a consequent activation of Th2 cytokines and increased production of IgE [14]. Additionally, there is increasing evidence that a balanced gut microbiome is needed for the proper formation of T-regulatory cells, which are important for tolerance induction [13]. 

### 1.3. Definitions Pro-, Pre- and Synbiotics

Probiotics are live microorganisms that, when administered in sufficient quantities, give a health improvement of the host. Probiotics induce immunomodulatory mechanisms in many different ways, including skewing of the Th1/Th2 balance towards Th1 by inhibiting Th2 cytokines or indirectly expanding IL-10 and Treg formation via either dendritic cell development or Toll-like receptors, although the exact mechanism remains to be clarified [15]. Prebiotics are substrates that are selectively utilized by host microorganisms conferring a health benefit. Synbiotics are defined as a mixture comprising live microorganisms and substrate(s) selectively utilized by host microorganisms that confer a health benefit on the host.

### 1.4. Rationale for Using Pro-, Pre- and Synbiotics in Atopic Diseases

Living circumstances in the industrialized world such as a decreased fermented food consumption, increased intake of antibiotics and other drugs, and improved hygiene are according to data from epidemiologic studies associated to the increase in allergic diseases. More or less exposure to microbial stimuli during infancy is associated to more or less allergic disease. The association has been described as the “hygiene hypothesis”. A lack of exposure to microbial stimuli early in childhood is a major factor involved in the steep increase in allergy [16]. In those who spend their childhood on a farm, allergic diseases are less common [16]. The comparison between the composition of microbiota of farm children and the microbiota of children with other lifestyles shows a significant difference [16]. Children living on farms are exposed to a wider range of microbes than children not living on a farm, and this exposure explains a substantial fraction of the inverse relation between asthma and growing up on a farm [17]. The gastrointestinal microbiota composition differs between allergic and healthy infants, independent of the prevalence of allergic disease in the region [13]. In contrary to what has been believed for a long time, an amniotic microbiome has been reported, and as a consequence, the fetal intestine may not be sterile since there is the presence of microbial deoxyribonucleic acid in meconium [18]. Early life is characterized by a rapid change in gastrointestinal microbiota composition. The first altering factor of the neonatal microbiome is the contact with vaginal, fecal and skin bacteria of the mother. In caesarean section-born babies, a less diversified microbiome is observed. The second altering factor is feeding. Human milk is rich in oligosaccharides which have prebiotic properties (a substrate that is selectively utilized by host microorganisms conferring a health benefit [19]) and promote the growth of selected species of bacteria. Human milk is also a natural bacterial inoculum. The third altering factor is environmental influenced alterations, which may undo the first two beneficial gut alterations: environments like neonatal intensive care units and medication such as antibiotics or proton pump inhibitors administered perinatal or during early life [20,21]. 

During early life, a balanced gastrointestinal microbiota is of major importance for the balanced skewing of the developing of the immune system and also determines the gut–lung communication of the gut–lung axis. Therefore, dysbiosis of the intestinal microbiome during early life will contribute to immune-mediated diseases later in life [14]. However, these associations between gut microbiota and allergic disease cannot provide a satisfactory explanation for all observations and does not result in evidence to decrease the rise in allergic disorders. However, the microbiota hypothesis does provide a rationale for using pro-, pre- and synbiotics, to alter the microbiota composition in the intestine to result in a more balanced development of the immune system [13]. Since a child’s microbiota does not reflect adult patterns until they are two years old, the infant microbiota may be more susceptible to manipulation [22]. 

More knowledge is needed on the mechanisms behind dysbiosis, translocation of microbiota from the intestine to the respiratory tract through various mechanisms and for a better evaluation of the therapeutic possibilities to correct this dysbiosis, which in turn can be used to manage various respiratory diseases [23]. 

In this paper, we will try to answer the question if probiotics or prebiotics and/or synbiotic supplementation can alter the microbiome sufficiently to have an efficacious prevention and/or management of allergic rhinitis and asthma. 

## 2. Materials and Methods

A search was performed in PubMed, EMBASE, Google Scholar, Web of Science and Cochrane Library. We included preferably meta-analyses, systematic reviews and clinical trials from 1990 up until October 2020 published in the English language. The following keywords in the respective language were used: “asthma”, “wheezing”, “respiratory disease”, “allergic rhinitis”, “allergic coryza”, “probiotics”, “prebiotics”, “synbiotics”, “prevention”, “therapy”, “therapeutics”, “child”. These keywords were combined with the Boolean command “OR” and were linked by the Boolean command “AND”. Records were screened based on the titles and abstracts. Articles were extracted using subject heading and key words of interest to the topic. A second selection was made by reading the abstract. Interesting references in included articles were also considered. Records were excluded if the abstract or full text was not available, if the topic was not relevant, if non-English or if the study design was not adequate. Duplicates were removed. 

Search strategy for human studies in the results section: In PubMed, the following search string was used: (“Asthma”[MeSH Terms] OR “respiratory disease”[Title/Abstract] OR “wheezing”[Title/Abstract] OR “recurrent wheeze”[Title/Abstract] OR “rhinitis, allergic, seasonal”[MeSH Terms] OR “allergic coryza”[Title/Abstract]) AND (“Probiotics”[MeSH Terms] OR “Prebiotics”[MeSH Terms] OR “Synbiotics”[MeSH Terms]) AND “Child”[MeSH Terms]. 

## 3. Results

### 3.1. Probiotics for Prevention of Asthma

#### 3.1.1. Animal Studies

A beneficial effect of the administration of probiotics was suggested by showing that oral administration of *Lactococcus lactis* NZ9000 to rats resulted in a decrease in infiltration of pro-inflammatory leucocytes, mainly eosinophils and decreased lung IL-4 and IL-5 expression in the broncho-alveolar lavage and a reduced level of serum allergen-specific IgE [24]. Another study conducted in mice using *Lactobacillus rhamnosus* GR-1 significantly prevented airway hyperreactivity development and prevented microbiome disturbance in the asthmatic animals, supporting the existence of the gut–lung axis [25]. An interesting aspect is that most probiotics are given orally; however, a new approach was tested by giving probiotics (*Lactobacillus paracasei* NCC2461 [26] and *Lactobacillus rhamnosus* GG [27] in mice through the nose and showed benefits in reducing inflammation of the lungs [28]. The probiotic *Bifidobacterium breve* M-16V administered to pregnant mice was shown to be effective in lowering eosinophils in the broncho-alveolar lavage fluid of neonatal mice and reduced allergic lung inflammation in mice exposed to air pollution [29]. In another animal study, the intranasal administration of *Lactobacillus rhamnosus* GG (LGG), but not *Lactobacillus rhamnosus* GR-1, suppressed airway hyper-reactivity and reduced the counts of eosinophils, IL-13 and IL-5 in broncho-alveolar fluid [27]. In addition to inhibiting inflammatory cell infiltration in lung tissue, *Lactobacillus* GG was shown to decrease MMP9 expression, a class of enzymes that are involved in the degradation of the extracellular matrix and of which levels were significantly increased in asthma [30]. *Lactobacillus* GG and *Bifidobacterium lactis* were shown to increase natural regulatory T cells in the lungs of asthmatic mice in another animal study [31]. Lee et al. mentioned that four *Lactobacillus* species used in animal studies had different immunomodulatory effects [32] against allergy *Lactobacillus planetarum* had shown some beneficial effect, but this was not the case for *Lactobacillus salivarius* and *fermentum* [33]. Probiotic strain-specific induction of Foxp3þ T regulatory cells was found in mouse allergy models [34].

#### 3.1.2. Human Studies 

In humans, evidence of the use of probiotics as a preventive agent for respiratory allergies in children was reported to be low [35] (Table 1). A meta-analysis of 2013 showed that by giving the most frequently used probiotics (*Lactobacillus* spp. and/or *Bifidobacteria* spp.) to prenatal mothers plus continued after birth versus only postnatally, no difference in IgE levels were seen. Less atopy was seen if the probiotics were given to pregnant women and continued after birth. Probiotics given after birth only decreases the risk of atopic sensitization in young children but not of asthma or wheeze [14]. This supports the theory that probiotics that have colonized the mothers’ intestine will be transferred at birth during vaginal delivery. Further administration of pro- and prebiotics to the pregnant mother results in the potential transmission of tolerogenic mediators such as regulatory cytokines, antibodies and growth factors across the placenta, stimulating the development of the fetal immune system [36]. This could help to prevent asthma or allergic rhinitis. Like mentioned above, the findings in pregnant mice are of human interest since up to now, knowledge was restricted to the fact that *Bifidobacterium breve* M-16V in infants can suppress T-helper type 2 immune responses and modulate the systemic Type 1 T helper/Type 2 T helper balance. Exposure of the pregnant mother to air pollution increases asthma susceptibility of the newborn and later on. Therefore, *Bifidobacterium breve* M-16V might contribute to reducing asthma in a population living in highly polluted areas [29]. In 2014, the Panda Study showed that giving a probiotic mixture postnatally (two *Bifidobactera* spp. and *Lactococcus lactis*) for one year does not have a beneficial effect on the development of allergic diseases after six years [37]. After five years follow-up, the negative outcome persisted [38]. Furthermore, no association (relative risk (RR) 0.59, 95% CI 0.36–0.96, *p* = 0.059) was found in a study with a follow-up of 11 years. This study was a two-center RCT using *Lactobacillus rhamnosus* HN001 or *Bifidobacterium lactis* HN019 daily taken from 35-week gestation to six months postpartum in mothers while breastfeeding and from birth to the age of two years in infants [39]. Consistent with the previously mentioned studies, a more recent meta-analysis including 19 RCTs involving 5157 children showed no association as well in lowering the incidence of asthma and wheezing if probiotics were given to pregnant mothers or postnatally. However, in infants with atopic diseases, probiotics seem to reduce the wheezing incidence significantly (RR 0.61, 95% CI 0.42–0.90; *p* < 0.05). No association was found between probiotics and a subgroup analysis of asthma (RR 0.94, 95% CI 0.82–1.09). Important to mention is that due to the small sample size in the subgroup analysis, the information should be interpreted carefully. The question “Do infants with atopic disease benefit from probiotics (Lactobacillus spp. and/or Bifidobacteria spp., Propionibacterium freudenreichii ssp. shermanii JS)?” should be tested in more heterogenetic, well-designed RCTs. Beneficial effects of specific strains might become lost by pooling probiotic strains together, since the effects are strain-specific. As a consequence, meta-analysis should be strain-specific. Due to the wide heterogeneity of strains, mixture and doses administered, the efficacy of specific probiotic strains has been difficult to analyze. Therefore, further research is needed to optimize the selection of probiotic strains and the configuration of intervention regimens [12].

### 3.2. Probiotics for the Treatment of Asthma 

The curative effects of probiotics in asthmatics are not well established [40] (Table 2). A recent study in baby mice indicated that *Bifidobacterium infantis* could reduce the infiltration of inflammatory cells by promoting Th1 immune responses and oppositely suppressing Th2 immune responses [41]. In a 2008 systematic review, probiotic administration showed no positive effect in the treatment of asthma [42]. A later meta-analysis from Das et al., which included 12 studies, showed no enhancement in quality-of-life scores in asthmatic patients. However, probiotics were found to be efficacious in diminishing the amount of asthma attacks [43]. Altogether, the present evidence does not support use of probiotics in the treatment of asthma, although some studies suggest some benefit while harm was not reported [40].

### 3.3. Probiotics for Prevention of Allergic Rhinitis

The occurrence of perennial allergic rhinitis and seasonal allergic rhinitis has been rising globally and their management is costly [44] (Table 3). Currently, there is no strong proof that probiotics are successful in preventing allergic rhinitis [45]. Surprisingly, some studies suggest that there may even be an increased prevalence of allergic rhino-conjunctivitis in patients taking probiotics in the perinatal period and in childhood [46]. In a systematic review published in 2014, five RCTs that have studied the preventive role of probiotics in allergic rhinitis were assessed. Combining data from adults and children, no difference in incidence of allergic rhinitis between the probiotic and control groups (odds ratio (OR) 1.07, 95% CI, 0.81–1.42, *p* = 0.64, fixed-effects model), and no significant difference in the prevention of allergic rhinitis have been found [47]. A 2019 meta-analysis of seventeen RCTs including 5264 children could not identify a clear advantage of probiotic supplementation during pre- and postnatal periods in the prevention of allergic rhinitis [48]. In follow-up research of a previous study investigating the pre- and postnatal usage of probiotics in high-risk children between five and ten years of age, Peldan et al. sent surveys to their parents to investigate if atopic diseases, including allergic rhinitis, were present. The lifetime prevalence of allergic rhinitis was equal in both probiotic and placebo groups (35.2% vs. 41.7%, adjusted OR 0.74, 95% CI 0.55–1.00, *p* < 0.05); nevertheless, the prevalence of allergic rhino-conjunctivitis at five to ten years of age was greater in the probiotic than in the placebo group (36.5% vs. 29.0%, OR 1.43, 95% CI 1.06–1.94, *p* = 0.03) [46]. Following the authors of this study, the question form may be biased since manifestations of viral rhinitis may be mistaken for allergic rhinitis [46]. After a follow-up of 11 years, the same negative outcome of no association (RR 0.85, 95% CI 0.65–1.1, *p* = 0.24) was found for probiotics *Lactobacillus rhamnosus* HN001 and *Bifidobacterium lactis HN019* taken by mothers every day from 35-week gestation to six months postnatally while breastfeeding and by infants from birth to two years of age [39]. However, similar to the prevention of asthma, the absence of evidence for a potential benefit may be due to shortcomings in study designs and the presence of multiple confounding variables. Probiotic intervention may have a favorable role in the prevention and additional treatment of allergic rhinitis, although results up to now are disappointing [49].

### 3.4. Probiotics for Treatment of Allergic Rhinitis 

Avoidance of contact with allergens, medications to reduce symptoms to decrease inflammation and immunotherapy are standard approaches in the management of allergic rhinitis [50]. The question raised is if oral probiotics might modulate the microbiome in such a way that they result in an alteration of the immune system which would contribute to the treatment of allergic rhinitis [51] (Table 4). The development of allergic inflammation in a murine house dust mite asthma model is suppressed by synbiotic mixtures of non-digestible oligosaccharides and *Bifidobacterium breve* M-16V [52]. 

A review from 2010 (including seven trials, n = 616, children and adults mixed) suggested that probiotics (*Lactobacillus* spp. and *Bifidobacterium* spp.) contribute to a decrease in allergic rhinitis symptoms, quality of life and decrease the need for drug intake (standard mean difference (SMD) −1.17, 95% CI −1.47–0.86; *p* < 0.00001) [53]. Another meta-analysis performed in 2014 including 11 RCTs reported similar conclusions, as probiotics significantly improved both quality of life and nasal symptom scores (SMD −2.97, 95% CI, −4.77–1.16, *p* = 0.001). However, this was not associated with an improvement in immunologic variables [47]. This meta-analysis was criticized for its methodology [47,54]. A 2016 meta-analysis of 22 RCTs also came up with evidence of a potential benefit of probiotics, once more demonstrating improvement in quality of life. A clinically significant benefit was reported for at least one outcome in 17 studies, while no benefit could be shown in six trials. Improvement was mainly regarding quality of life (SMD −2.30, 85% CI −3.93 to −0.67, *p* = 0.006), while no effect was shown on rhinitis symptoms (SMD −0.34, 95% CI −0.62–0.07; *p* = 0.13) or total IgE levels (SMD 0.01, 95% CI −0.17–0.19, *p* = 0.88), and for antigen-specific IgE (SMD 0.09, 95% CI −0.44–0.62, *p* = 0.74) in the placebo group compared to the probiotic. Studies are characterized by a high degree of heterogeneity in probiotic strains tested, inclusion criteria and outcomes [55]. 

In 212 children under five-years-old from Pakistan, a probiotic product administered as a chewable tablet, containing two x 10^9^ CFU of *Lactobacillus* Paracasei (LP-33), was administered once a day for six weeks while the control group was treated with cetirizine tablet 2.5 mg (<two years) or 5 mg (two-five years) once daily. Significant improvement from baseline symptoms (rhinorrhea, sneezing, nasal blocking, coughing, feeding difficulties and sleeping difficulties) was reported equally in both groups in almost all children [56]. Although the title of the paper mentions probiotics, the study was in fact performed with postbiotics since it was lyophilized extracts of bifidobacteria which were shown to suppress allergic rhinitis in mice via inducing IL-10-producing B cells [57]. Another study (with mice) showed that Clostridium butyricum extracts—again, postbiotics—can efficiently inhibit experimental allergic rhinitis by increasing IL-10 expression in B cells [58].

A pilot study in only 20 adult (18–65-years-old) patients with allergic rhinitis caused by house dust mite allergy suggests that probiotics-impregnated bed linen with five natural genetically unmodified bacterial probiotic strains of Bacillus species (strains of Bacillus subtilis, Bacillus amyloliquefaciens and Bacillus pumilus) reduces symptoms and increases quality of life [59]. A large-scale study is recommended to further investigate all these findings [59].

### 3.5. Prebiotics for Prevention/Treatment of Asthma or Allergic Rhinitis 

Inulin, fructo-oligosaccharides and galacto-oligosaccharides are well known examples of prebiotics. Table 5 provides an overview of the literature. These substrates will contribute to the growth of two common bacteria in the gut-bifidobacteria and lactobacilli [60]. Some of the substrates interacting with the infant’s gut microbiome are human milk oligosaccharides (HMOs) [61], which form the third biggest fraction in human milk [36]. In a mouse model, 2’-fucosyllactose and 6’-sialyllactose decrease the symptoms of food allergy due to the induction of IL-10(+) T regulatory cells and indirect stabilization of mast cells [62]. Prebiotics such as non-human galacto- and fructo-oligosaccharides have been added to infant formula to try to mimic the results of HMOs. However, these non-human prebiotics are less structurally diverse than HMOs [47]. An 18-year follow-up of high-allergy-risk breastfed infants was conducted to evaluate the relation between HMO profiles of the mother and the risk of developing asthma, eczema and sensitization. One HMO profile, namely the acidic Lewis HMOs, showed an increased risk of developing allergic disease and asthma in youth (OR 5.82, 95% CI 1.59–21.23) compared to the neutral Lewis HMO profile. Another finding of the study is that the acidic-predominant profile was associated with a lower risk of food sensitization (OR 0.08, 95% CI 0.01–0.67, *p* < 0.05). HMOs have only been recently available on the market; nevertheless, there are some studies investigating their effect on allergies [63]. A meta-analysis with two studies reporting early respiratory symptoms as outcome (n = 249) has examined if these non-human oligosaccharides have effects on allergy. The study found that infants who received prebiotics (non-human oligosaccharides) had reduced asthma or recurrent wheezing (RR 0.37, 95% CI 0.17–0.80, *p* < 0.01) [64]. Another double blinded RCT (n = 461) compared Chinese toddlers drinking standard milk formula with those drinking a formula containing bioactive proteins and/or the HMO 2′-fucosyllactose and/or milk fat, for a period of six months. In this study, however, no difference was found in the occurrence of upper respiratory infections. No analysis for allergy was conducted [65]. Concluding, there is still little evidence to use prebiotics for the prevention of asthma and none for allergic rhinitis to our knowledge on rhinitis. No studies have been conducted to analyze the effects of prebiotics as a treatment for asthma or allergic rhinitis.

### 3.6. Synbiotics for Prevention/Treatment of Asthma or Allergic Rhinitis

#### 3.6.1. Asthma 

The literature on synbiotics regarding prevention and/or treatment of allergic manifestations is still limited (Table 6). Some analyses do not differentiate between pre-, pro- and synbiotics. [66]. Ninety infants with atopic dermatitis were managed with a formula with extensively hydrolyzed protein and were included in a double-blind, placebo controlled multicenter trial for 12 weeks, randomized to the formula with or without synbiotics over a period of seven months. One year later, information regarding respiratory symptoms and asthma medication was collected with a questionnaire. The significant reduced prevalence of “frequent wheezing” and “wheezing and/or noisy breathing apart from colds” was observed in the synbiotic group (13.9% vs. 34.2%, absolute risk reduction (ARR) −20.3%, 95% CI −39.2% to −1.5%, and 2.8% vs. 30.8%, ARR −28.0%, 95% CI −43.3% to −12.5%, respectively). Additionally, the use of asthma medication was significantly lower (5.6% vs. 25.6%, ARR −20.1%, 95% CI −35.7% to −4.5%). However, total IgE levels did not differ. Increased specific cat-IgE levels were noticed in five children (15.2%) in the placebo group versus none in the synbiotic group (ARR −15.2%, 95% CI −27.4% to −2.9%). The outcome of this trial suggests that synbiotics may prevent asthma in infants presenting with atopic dermatitis [67]. However, the limited number of children included in this trial is a major limitation. Cabana et al. [68] performed an RCT in 92 infants with a mixture of LGG and inulin as synbiotic (in the study mentioned as probiotics) between birth and the age of six months of life in infants with mixed breast and formula feeding [68]. Asthma at the age of five years was a secondary outcome, but was not statistically different in both groups with an incidence of 17.4% in the control prebiotic and 9.7% in the symbiotic [68].

A double-blinded, placebo-controlled RCT performed in Iranian children younger than 12 years tested the efficacy of synbiotic (Kidilact^®^: Streptococcus thermophilus, *Bifidobacterium* spp., *Lactobacillus* spp. zinc and fructo-oligosaccharide) asthma management. Multiple outcomes did not show a difference between both groups; the number of outpatient visits, 19 in the synbiotic versus 55 in the control arm (*p* = 0.001), was the only statistically significant difference [69].

#### 3.6.2. Allergic Rhinitis 

The effect of synbiotics on prevention of allergic rhinitis will remain unanswered because no RCTs have been conducted yet to our knowledge (Table 7). Clinical symptoms and quality of life improve with immunotherapy, but synbiotics do not contribute to this improvement. 

Synbiotics in the treatment of allergic rhinitis are also poorly studied, although some of the trials reporting on the efficacy of probiotics, in fact, concern synbiotics [37]. A placebo-controlled, double-blind RCT in a small number of children and adults (*n* = 20, age nine-53 years) in Iran showed that immunotherapy and a synbiotic (Streptococcus thermophilus, *Bifidobacterium* spp., *Lactobacillus* spp., fructo-oligosaccharide) reduced the gene expression of IL-17 after two and six months (*p* = 0.001, *p* = 0.0001) more compared to the group receiving immunotherapy and a placebo [70]. Other probiotics [71] were also shown to reduce cytokine IL-17 by directly and indirectly downregulating and suppressing the T helper 17 subset. A 2019 crossover RCT (*n* = 152 subjects (30.1 ± 7.6 years) in adults in Iran showed that adding synbiotics (however, in the study, mentioned as probiotics) to budesonide significantly ameliorated quality of life in persistent allergic rhinitis patients (*p* < 0.05 for social functioning and *p* < 0.001 for mental health and vitalism) [37]. The patient population used in this study may not be representative for allergic rhinitis patients in the overall population, since symptoms did not respond to usual therapy with antihistamines, antileukotrienes, decongestants and nasal steroids [51]. 

More well-designed studies, investigating only the effects of synbiotics for allergy prevention and/or treatment, are needed [36].

## 4. Conclusions

Meta-analyses have showed marked heterogeneity as well in inclusion criteria, studied products and primary outcomes between studies, making direct comparison hazardous. Today, the American Academy of Pediatrics, the European Academy of Allergy and Clinical Immunology, the National Institute of Allergy and Infectious Disease, and the European Society for Pediatric Gastroenterology, Hepatology and Nutrition do not recommend the use of probiotics for primary prevention of allergic disease [13]. The lack of evidence is the consequence of large heterogeneities between study designs, differences in strains, and dosages and duration of probiotics administered. Future research may clarify these issues [35]. Data from laboratory research in well-controlled conditions demonstrate an important role for gastrointestinal microbiota composition on the development of allergic disease in the respiratory tract, suggesting even a causal relation. Data from clinical human studies remain disappointing. The multiple confounding variables in the clinical situation, therefore, illustrate the impact of environmental and other variables on the development of allergic disease. Overall, we have to conclude that the evidence is insufficient to recommend administration of pro-, pre- or synbiotics in the prevention or treatment of respiratory tract allergies. However, adverse effects are not reported. Additionally, data obtained in controlled situations suggest benefits. Future research requires thoughtful development of appropriate study design according to internationally set standards to ensure uniformity [72]. The modes of action of pro-, pre- and synbiotics need to be further clarified in health and disease [40].

## Figures and Tables

**Table 1 nutrients-13-00934-t001:** Studies examining probiotics for prevention of asthma in humans.

#	Author and Publication Date	Country	Type of Study	Number of Trials and or Patients	Age	Type of Probiotic and Dose(s) (cfu)	Administration Duration of Probiotics	Follow-Up	Effect
1	Elazab et al., 2013 [14]	United States of America	Metanalysis including: Double-blinded, randomized, placebo-controlled trials	25 studies of 20 cohorts (*n* = 4031) *Only* 10 trials were included (*n* = 3143) to look at probiotics and risk of asthma/wheeze	Birth up to 6 years.	*Lactobacillus* spp. and *Bifidobacterium* spp. or mixed probiotics *Dose:* 1–550 × 10^8^ cfu	1–13.5 months 8 trial probiotics administrated also prenatal	0–70 months	Probiotics did not significantly reduce asthma/wheeze (RR 0.96, 95% CI 0.85–1.07), no evidence of publication bias (*p* = 0.25)
2	Gorissen et al., 2014 [37]	The Netherlands	Prospectively in a single-blinded (investigator blinded) design.	*n* = 123 and 83 at age of 6 years	Birth up to 6 years	Probiotic mixture consisting of *B bifidum*, *B lactis* and *Lactococcus lactis* *Dose:* not mentioned	0–24 months Administration started prenatal.	0–24 months and once at 6 years of age	Did not lead to prevention of asthma at 1 and 6 years of age
3	Wickens et al. 2018 [39]	New Zealand	two-center randomized placebo-controlled trial	*n* = 407	Birth–11 years	*L. rhamnosus* HN001 or *B. lactis* HN019 *Dose*: 6 × 10^9^ cfu or 9 × 10^9^ cfu	Daily from 35-week gestation to 6 months’ post-partum in mothers while breastfeeding and birth to age 1 years in infants	Birth to 11 years	No association with the development of allergic disease was found RR 0.59, 95% CI 0.36–0.96, *p* = 0.059
4	Wei et al., 2020 [12]	China	meta-analysis included randomized, double blind, placebo-controlled trials	Total: 19 RCTs (*n* = 5157) 14 RCTs *n* = 4021 for the analysis of asthma (of which 10 RCTs already used in Elazab et al. (2013))	Birth- 8 years	10 RCTs *L.* spp. 1 RCT *B.* spp. 6 RCTs probiotic mixtures *Dose*: daily ranged from 10^8^ to 10^11^ cfu	3–24 months	1–8 years	No significant association of probiotics with risk of asthma (RR 0.94, 95% CI, 0.82–1.09) or wheeze (RR 0.97, 95% CI, 0.88–1.06) compared with placebo. Subgroup analysis by asthma risk showed that probiotics significantly reduced wheeze incidence among infants with atopy disease (RR 0.61, 95% CI, 0.42–0.90),
5	Davies et al., 2018 [38]	United Kingdom	randomized, double-blind, placebo-controlled, parallel group trial	*n* = 318	Birth up to 5 years	*L.* spp. and *B.* spp. or mixed probiotics *Dose*: 1 × 10^10^ cfu per day	from 36 weeks’ gestation, and then administered to their infants during their first 6 months of life	Follow up at 2 years and 5 years	No reduction in asthma after 2 or 5 years

Legend. #= number; RCT, randomized controlled trial; spp., species; cfu, colony-forming unit; CI, confidence interval; RR, relative risk; L: Lactobacillus; B: Bifidobacterium.

**Table 2 nutrients-13-00934-t002:** Studies examining probiotics for the treatment of asthma.

#	Author and Publication Date	Country	Type of Study	Number of Patients	Age	Type of Probiotic	Administration Duration of Probiotics	Follow-Up	Effect
1	Sharma et al., 2018 [40]	Korea	Review	1 RCT examining probiotics for treatment of asthma with *n* = 105	6 up to 2 years	*L. gasseri* PM-A0005 (A5; 2 × 10^9^cells/capsule) twice a day	8 weeks	Observation period of 10 weeks	Pulmonary function and PEFR increased significantly and the clinical symptom scores for asthma decreased in the probiotic group
2	Vliagoftis et al., 2008 [42]	Multi-center (Greece, The United States of America, Canada)	Systematic review of randomized controlled trials	4 RCTs (*n*= 257)	2–13 years (1 RCT included patients up to 45 years)	1 RCT *L. casei* (10^10^ cfu); 1 RCT *Enterococcus faecalis* (18 × 10^7^ cfu); 1 RCT *L. rhamnosus* (10^10^ cfu); 1 RCT *L.acidophilus* (7.6 × 10^8^ cfu)	4 weeks up to 1 year	22 up to 56 weeks	No effect of probiotics on asthma treatment
3	Das et al., 2013 [43]	India	Systematic review	10 RCTs (*n* = 860) and 2 Randomized crossover designs (*n* = 39)	2 up to 16 years (7 RCTs also included adults up to 57 years)	Different strains (*L. salivarius, gasseri, acidophilus, paracasei, rhamnosus, Bulgaricus; Streptococcus thermophilus; B. longum* 5 36)	1 month up to 1 year	Not mentioned	No improvement in quality-of-life score in asthmatics. Longer time free from episodes of asthma (mean (95% CI 3.5, 2.7–4.3) versus 2.1 (1.5–2.7) months) (*p* = 0.027))

Legend: #= number; RCT, randomized controlled trial; spp., species; cfu, colony-forming unit; CI, confidence interval; RR, relative risk; L: Lactobacillus; B: Bifidobacterium.

**Table 3 nutrients-13-00934-t003:** Studies examining probiotics for prevention of allergic rhinitis.

#	Author and Publication Date	Country	Type of Study	Number of Patients	Age	Type of Probiotic	Administration Duration of Probiotics	Follow-up	Effect
1	Zuccotti et al., 2015 [45]	Italy	Systematic review and meta-analysis	17 RCTs (*n* = 4755)	Children, not otherwise specified	##	##	2 months up to 7 years	No significant difference in terms of prevention of rhino-conjunctivitis (RR 0.91, 95% CI 0.67–1.23, *p* = 0.53) was documented.
2	Peng et al., 2015 [47]	China	Systematic review	A total of 11 RCTs of which 5 addressed the preventive role of probiotics in AR (*n* = 1527)	Mothers from 36 weeks of gestation; Infants from birth up to adults	*L.* spp. and *B.* spp. or mixed probiotics *Dose*: wide range of probiotic doses applied	##	##	No difference in the incidence of AR between probiotic and placebo groups. Improvement in overall quality of life and nasal symptom scores (MD—2.97 95% CI, −4.77–1.16; *p* = 0.001).
3	Du et al., 2019 [48]	China	Meta-analysis	17 RCTs (*n* = 5264)	Children	Variable strains	##	##	No clear benefit of probiotics in the prevention of allergic rhinitis
4	Wickens et al., 2018 [39]	New-Zealand	Two-center randomized placebo-controlled trial	*n* = 407	Birth up to 11 years	*L. rhamnosus* HN001 or *B. lactis* HN019 *Dose*: 6 × 10^9^ cfu or 9 × 10^9^ cfu	Daily from 35-week gestation to 6 months post-partum in mothers while breastfeeding and birth to age 1 years in infants	Birth up to 11 years	Children taking HN001 had a notable reduction in the risk of rhinitis (RR 0.79, 95% CI 0.59–1.05, *p* = 0.1). Among *Bifidobacterium lactis* HN019 children, there was no notable reduction in allergic rhinitis prevalence.

Legend: #= number; RCT, randomized controlled trials; spp., species; cfu, colony-forming unit; CI, conventional interval; RR, relative risk, ## not mentioned or not applicable; L: Lactobacillus; B: Bifidobacterium

**Table 4 nutrients-13-00934-t004:** Studies examining probiotics for treatment of allergic rhinitis.

#	Author and Publication Date	Country	Type of Study	Number of Patients	Age	Type of Probiotic	Administration Duration of Probiotics	Long-Term Follow Up	Effect
1	Das et al., 2010 [53]	India	systematic reviews	7 RCTs *n* = 616	Any age	*L.* spp. and *B.* spp. Additionally, one RCT Streptococcus thermophilus *Dose:* Huge variation	1 up to 2 months	none	Decrease in allergic rhinitis symptoms, quality of life and need for drug intake
2	Peng et al., 2015[47]	China	systematic review and meta-analysis	11 RCTs *n* = 1527	During pregnancy up to 9 months of age	Different: *Propionibacterium freudenreichii* ssp. shermanii JS, *S. thermophiles; L.* spp. and *B.* spp. *Dose*: Huge variation	4–9 months Some started during pregnancy	none	Significantly improved both quality of life and nasal symptom scores
3	Guvenc et al., 2016 [55]	Turkey	Systematic review and metanalysis	22 RCTs *n* = 2242 (*n* = 1953 after losses to follow-up)	2 up to 65 years of age	Huge variance in probiotics and doses	From 1 up to 12 months	none	16 RCTs had significant benefits of probiotics on clinical parameters; 9 RCTs had significant improvement in immunologic parameters compared with placebo. Meta-analysis significant ameliorated nasal and ocular symptoms and QoL scores
4	Ahmed et al., 2019 [56]	Pakistan	RCT	*n* = 212	6 to 60 months	*L. Paracasei* (LP-33) *Doses*: 2 × 10^9^ cfu once daily	6 weeks	none	Probiotic (LP-33) was equally effective as cetirizine in under five years children for the treatment of perennial allergic rhinitis
5	Berings et al., 2017 [59]	Belgium	pilot double-blind, randomized, placebo-controlled, crossover trial	*n* = 24	18–65 years	Purotex^®^ textile treatment contains five different probiotic and natural (not genetically modified) bacterial strains of *Bacillus* species. *Doses*: unknown			significant improvement in symptoms and QoL

Legend. #: number; RCT randomized controlled trials, spp. species cfu: colony-forming unit, CI confidence interval, RR relative risk; L: Lactobacillus; B: Bifidobacterium; S: Stretococcus.

**Table 5 nutrients-13-00934-t005:** Studies examining prebiotics for prevention/treatment of asthma or allergic rhinitis.

#	Author and Publication Date	Country	Type of Study	Number of Patients	Age	Type of Prebiotic	Administration Duration of Prebiotics	Long-Term Follow Up	Effect
1	Lodge et al., 2020 [63]	Australia	randomized controlled trial of the effects of infant formulas weaning on allergic disease risk, then continued as an observational birth cohort	*n* = 145	0–18 years	HMO	0–12 months breastfeeding	18 times in the first 2 years, then yearly until 7 years, then at 12 and 18 years	some profiles of HMOs were associated with increased and some with decreased allergic disease risks over childhood
2	Cuello-Garcia et al. 2016 [64]	International ??	Meta-analysis 2 RCTs	*n* = 249	Infant	HMO	##	##	HMO reduced asthma or recurrent wheezing
3	Leung et al., 2020 [65]	China	randomized, controlled, double-blind, parallel-group clinical trial	*n* = 461	1–2.5 years	standard formula milk or containing bioactive proteins and/or the HMO 2′-fucosyllactose and/or milk fat	6 months	6 months	No reduction for respiratory and gastrointestinal infections in toddlers with HMO

Legend. #: number; HMO, human milk oligosaccharide; RCT, randomized controlled trial; spp., species; cfu, colony-forming unit; CI, confidence interval; RR, relative risk; ## not mentioned or not applicable.

**Table 6 nutrients-13-00934-t006:** Studies examining synbiotics for prevention/treatment of asthma.

#	Author and Publication Date	Country	Type of Study	Number of Patients	Age	Type of Synbiotic	Administration Duration of Syniotics	Long-Term Follow Up	Effect
1	van der Aa et al., 2011 [67]	The Netherlands	double-blind, placebo-controlled multicenter trial	*n* = 90 follow up *n* = 75	mean age 17.3 months	extensively hydrolyzed formula with *B. breve* M-16V and a galacto/fructo-oligosaccharide mixture *Dose*: Not mentioned	4 weeks	1 year	synbiotic mixture prevents asthma-like symptoms in infants with atopic dermatitis. Additionally, less started with medication
2	Cabana et al., 2017 [68]	United states of America	randomized, double-blind controlled trial	*n* = 92	4 days old–6 years	10^10^ cfu *L. rhamnosus* GG and 225 mg of inulin for first 6 months of life	6 months	Up to 6 years	No significant reduction in asthma with synbiotics
3	Hassanzad et al., 2019 [69]	Iran	double-blinded, randomized, placebo-controlled clinical trial	*n* = 100	6.9 ± 2.7 years	*S. thermophiles; L.* spp. and *B.* spp., zinc and FOS (prebiotic)	6 months	Not mentioned	Less outpatient visits, no significant frequency of asthma attacks and hospitalization due to asthma being exacerbated

Legend. #:number; RCT, randomized controlled trial; spp., species; cfu, colony-forming unit; CI, confidence interval; RR, relative risk; L: Lactobacillus; B: Bifidobacterium; S: Streptococcus; FOS: fructo-oligosaccharide.

**Table 7 nutrients-13-00934-t007:** Studies examining synbiotics for prevention/treatment of allergic rhinitis.

#	Author and Publication Date	Country	Type of Study	Number of Patients	Age	Type of Synbiotic	Administration Duration of Synbtics	Long-Term Follow Up	Effect
1	Dehnavi et al., 2019 [70]	Iran	placebo-controlled, double-blind RCT	*n* = 20	9 up to 53 years	*S. thermophilus*, *B.* spp., *L.* spp., FOS *Doses*: not written	2 months	Total of 6 months	Significant reduction in IL-17 gene expression following administration of symbiotic. Clinical symptoms and quality of life were improved with immunotherapy. Synbiotics did not have additional effects
2	Jalali et al., 2019 [51]	Iran	Crossover RCT	*n* = 152	Adults; 30.1 ± 7.6 years	seven different Gram-positive organisms: 9 × 10^9^ cfu/g lyophilized lactobacilli (*L. acidophilus*, *L. casei*, *L. delbrueckii* subsp. L. bulgaricus, and *L. rhamnosus*), 1.25 × 10^10^ of bifidobacteria (*B. longum*, and *B. breve*), and 1.5 × 10^−10^ of *S. salivarius* subsp. thermophilus and 38.5 mg FOS	4 months	Total 2 months	Addition of probiotics to budesonide significantly improved QoL in persistent AR patients

Legend. #: number; RCT, randomized controlled trial; spp., species; cfu, colony-forming unit; CI, confidence interval; RR, relative risk; L: Lactobacillus; B: Bifidobacterium; S; Stretococcus; FOS; fructo-oligosaccharide.

## Data Availability

Not applicable.

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
