# Peer review of "Prevention and Management with Pro-, Pre and Synbiotics in Children with Asthma and Allergic Rhinitis: A Narrative Review"

_nutrients, 2021, doi:10.3390/nu13030934_

Round 1

Reviewer 1 Report

The paper “Prevention and management with pro-, pre and synbiotics in children with asthma and allergic rhinitis: a narrative review” focuses on the potential role of pro-, pre and synbiotics in allergic diseases prevention and treatment. The paper is overall well written and focuses on a hot topic. This is a narrative review, however, considering that the Authors searched the literature including papers published from 1990 to 2020, the presentation of the relevant studies seems too synthetic and the methodology of the search itself is not sufficiently described (which kind of study was included? Which were the exclusion criteria? Which keywords were used?). I think the Authors should review the paper in order to check if all the most important pediatric studies and RCT published so far heve been reported and discussed, and add tables summarizing their findings for each section.

Minor issues:

- Page 3, lines 96-97: As far as children living in farms and related risk of allergy, I suggest to cite an important paper by Ege MJ et al. (Exposure to environmental microorganisms and childhood asthma. N Engl J Med. 2011 Feb 24;364(8):701-9)

- Page 3, line 143-144: this sentence seems redundant, since these concepts have been stated elsewhere in the text.

- Page 4, line 155: “the nasal” -> “the nose”; line 174 “ the frequent” -> “the most frequently used”

- Page 5 lines 214-221: Authors should clarify whether these studies were performed in adults or in children.

- Page 5, line 226 “suggesting” -> “suggest”

- Page 6 lines 248-249: the sentence on chronic rhinosinusitis seems too synthetic and off topic  / lines 258-260: it is not clear what the Authors mean, and what data do they refer to.

- Page 7 line 330: what do “extensive” refer to?

Author Response

Dear Editor

We are pleased to submit the revised version of the paper "Prevention and management with pro-, pre and synbiotics in children with asthma and allergic rhinitis: a narrative review."

The vast majority of the excellent comments and suggestions of the reviewers were included in the revised version.

We apologize for the delayed re-submission but the shared first author of this paper (Lien Meirlaen) delivered a preterm baby with serious health problems.

Thanks for your understanding.

Yvan Vandenplas

Reviewer 1

We thank the reviewer for the constructive comments and suggestions. We tried to include all of these recommendations in the revised paper.

This is a narrative review, however, considering that the Authors searched the literature including papers published from 1990 to 2020, the presentation of the relevant studies seems too synthetic and the methodology of the search itself is not sufficiently described (which kind of study was included? Which were the exclusion criteria? Which keywords were used?). I think the Authors should review the paper in order to check if all the most important pediatric studies and RCT published so far have been reported and discussed, and add tables summarizing their findings for each section.

Methodology further described with description of included type of studies (L139-140) “We included preferably meta-analyses, systematic reviews and clinical trials” and list of keywords “The following keywords in the respective language were used: “asthma”, “wheezing”, “respiratory disease”, “allergic rhinitis”, “allergic coryza”, “probiotics”, “prebiotics”, “synbiotics”, “prevention”, “therapy” and “therapeutics”.”

You can find our revision of literature with tables summarizing our findings for each section at the end of this paper

Page 3, lines 96-97: As far as children living in farms and related risk of allergy, I suggest to cite an important paper by Ege MJ et al. (Exposure to environmental microorganisms and childhood asthma. N Engl J Med. 2011 Feb 24;364(8):701-9)

We thank the reviewer for the excellent suggestion. The reference was added to the manuscript.

Page 3, line 143-144: this sentence seems redundant, since these concepts have been stated elsewhere in the text.

We deleted the sentence here, but did end the introduction section on the prevalence and burden caused by asthma with this statement.

Page 4, line 155: “the nasal” -> “the nose”; line 174 “ the frequent” -> “the most frequently used”

Thanks to the reviewer for having noticed these typo's. They were corrected.

Page 5 lines 214-221: Authors should clarify whether these studies were performed in adults or in children.

The requested information was added

Page 5, line 226 “suggesting” -> “suggest”

Adapted

Page 6 lines 248-249: the sentence on chronic rhinosinusitis seems too synthetic and off topic

The sentence was deleted.

lines 258-260: it is not clear what the Authors mean, and what data do they refer to.

We changed the sentence, and do hope it is more clear now "The question raised is if oral probiotics might modulate the microbiome in   such a way that they result in an alteration of the  immune system which would  contribute to the treatment of allergic rhinitis (51)".

Page 7 line 330: what do “extensive” refer to?

This has been adapted: "formula with extensively hydrolyzed protein"

Reviewer 2 Report

This review article refers to allergic diseases prevention and treatment using pro-, pre-, and synbiotics. The problem is essential and is now widely discussed.

I have some comments, which are stated below.

Introduction - 1.2. Pathophysiology asthma and allergic rhinitis

That section should be shortened and limited to work-related issues, e.g., the sentence: “Continuous exposure to allergens and supplemental triggering factors (eg. infections, tobacco use, pollution and physical exercise) cause chronic inflammation of the mucosa and submucosa, which leads to mucosal remodeling, possible irreversible tissue damage that contributes to the severity of the disease.” – is irrelevant and should be omitted, similarly to the descriptions of the diseases’ symptoms.  Authors need to focus on the possible modulatory role of probiotic interventions.

Line 75: “One should not forget…..” – please omit this part of the sentence

Line 86: The sentence “Little is known of the second group and the third group is a mixed group of the first two groups, sometimes leading to an inflammatory profile over time (5)” -  unclear; please clarify what a second and third group is.

In the Introduction section, one more chapter would be useful, defining pre- pro- and synbiotics with their likely impact on food allergy. It would not be necessary to provide their description once again in the other parts of the manuscript.

Results

The layout of the manuscript in the Results section is a bit confusing. I propose to characterize all animal studies first (e.g., lines 212-216: mice study and then humans are in the same chapter and even in the same line), followed by human studies divided into the prevention of respiratory tract allergic diseases, then impact on the treatment of allergic rhinitis followed by the asthma therapy, possibly with subsections into pre-, pro- and synbiotics, if appropriate. The clinical division is more critical than the division into pro-, pre- and synbiotics.

Line 199: “However, in infants with atopic diseases, probiotics seem to reduce the wheezing incidence significantly following a subgroup analysis by asthma risk (RR 0.61, 95% CI, 0.42–0.90)” – please provide a p-value; a confidence interval might suggest that decline in RR is insignificant 

The same comment line 310: ” Another finding of the study is that the acidic-predominant profile was associated with a lower risk of food sensitization (OR 0.08, 95% CI 0.01-0.67) – please provide a p-value

The same comment: line 316 and 327

Line 237: “ The lifetime prevalence of allergic rhinitis was equal in both probiotic and placebo groups; nevertheless, the prevalence of allergic rhino conjunctivitis at five-ten years of age was greater in the probiotic than in the placebo group” – please provide RR or OR

Line 259: “This paper discusses in fact synbiotic administration and not probiotic as mentioned in the title (51)” – which paper, please clarified, a sentence is not related to the previous context

In turn, previous sentences: “First-line management consists of corticosteroid sprays administered intranasal (50). However, these medications have often adverse effects with a negative impact on the quality of life (49). As a consequence, patient compliance is a main concern” – are not related to the paper issue and, in my opinion, should be omitted

Line 260: “A review  from 2010 (including seven trails, n=616, children and adults mixed) suggested that probiotics (Lactobacillus spp and Bifidobacterium spp) contribute to a decrease of allergic rhinitis symptoms, quality of life and decrease the need for drug intake (52).” – please provide pooled RR or HR with CIs and pooled p-values.

Line 265 – SMD – please explain the abbreviation and provide CIs and p-value for all SMD; otherwise they are unconfident

Line 343: Cabana et al…. – please provide ref No after the author’s name

Line 367: “A 2019 crossover RCT (n=152 subjects (30.1 ± 7.6 years) in adults showed that adding synbiotics to budesonide significantly ameliorated quality of life in persistent allergic rhinitis patients (37) – please provide RR, CI, and a p-value. 

And the next sentence: “The patient population used in this study may be not very representative of allergic rhinitis patients in the overall population (51)” – why, please clarify – unclear.

Author Response

Dear Editor

We are pleased to submit the revised version of the paper "Prevention and management with pro-, pre and synbiotics in children with asthma and allergic rhinitis: a narrative review."

The vast majority of the excellent comments and suggestions of the reviewers were included in the revised version.

We apologize for the delayed re-submission but the shared first author of this paper (Lien Meirlaen) delivered a preterm baby with serious health problems.

Thanks for your understanding.

Yvan Vandenplas

Reviewer 2

Reviewer 2

Introduction -

1.2. Pathophysiology asthma and allergic rhinitis

That section should be shortened and limited to work-related issues, e.g., the sentence: “Continuous exposure to allergens and supplemental triggering factors (eg. infections, tobacco use, pollution and physical exercise) cause chronic inflammation of the mucosa and submucosa, which leads to mucosal remodeling, possible irreversible tissue damage that contributes to the severity of the disease.” – is irrelevant and should be omitted, similarly to the descriptions of the diseases’ symptoms.  Authors need to focus on the possible modulatory role of probiotic interventions.

The statement was deleted.

Line 75: “One should not forget…..” – please omit this part of the sentence

This was deleted.

Line 86: The sentence “Little is known of the second group and the third group is a mixed group of the first two groups, sometimes leading to an inflammatory profile over time (5)” -  unclear; please clarify what a second and third group is.

We agree that the sentence is confusing, and does in fact not really add valuable information. Therefore, the statement was deleted.

In the Introduction section, one more chapter would be useful, defining pre- pro- and synbiotics with their likely impact on food allergy. It would not be necessary to provide their description once again in the other parts of the manuscript.

A paragraph with the requested information was added (1.3 section)

Results section

The layout of the manuscript in the Results section is a bit confusing. I propose to characterize all animal studies first (e.g., lines 212-216: mice study and then humans are in the same chapter and even in the same line), followed by human studies divided into the prevention of respiratory tract allergic diseases, then impact on the treatment of allergic rhinitis followed by the asthma therapy, possibly with subsections into pre-, pro- and synbiotics, if appropriate. The clinical division is more critical than the division into pro-, pre- and synbiotics.

The reviewer will have noticed that we did use the proposed approach for the prevention of asthma. We voluntary decided to do different for the section on treatment because the animal studies do underline the mechanisms of action and explain the results observed in the human studies. The structure we used shows the results in human studies and then explains the "why" with data from animal studies. Mixing data obtained with pro-, pre- and synbiotic interventions would also be quite confusing for the reader.

Line 199: “However, in infants with atopic diseases, probiotics seem to reduce the wheezing incidence significantly following a subgroup analysis by asthma risk (RR 0.61, 95% CI, 0.42–0.90)” – please provide a p-value; a confidence interval might suggest that decline in RR is insignificant

P value was <0.05. This was added to the paper.

The same comment line 310: ” Another finding of the study is that the acidic-predominant profile was associated with a lower risk of food sensitization (OR 0.08, 95% CI 0.01-0.67) – please provide a p-value

Also here the p value was <0.05

The same comment: line 316 and 327

Line 316: p <0.01. Line 327 was deleted.

Line 237: “ The lifetime prevalence of allergic rhinitis was equal in both probiotic and placebo groups; nevertheless, the prevalence of allergic rhino conjunctivitis at five-ten years of age was greater in the probiotic than in the placebo group” – please provide RR or OR

The requested information was added to the manuscript?

Line 259: “This paper discusses in fact synbiotic administration and not probiotic as mentioned in the title (51)” – which paper, please clarified, a sentence is not related to the previous context

The sentence was rephrased.

In turn, previous sentences: “First-line management consists of corticosteroid sprays administered intranasal (50). However, these medications have often adverse effects with a negative impact on the quality of life (49). As a consequence, patient compliance is a main concern” – are not related to the paper issue and, in my opinion, should be omitted.

We deleted these sentences

Line 260: “A review  from 2010 (including seven trails, n=616, children and adults mixed) suggested that probiotics (Lactobacillus spp and Bifidobacterium spp) contribute to a decrease of allergic rhinitis symptoms, quality of life and decrease the need for drug intake (52).” – please provide pooled RR or HR with CIs and pooled p-values.

Requested information was added.

Line 265 – SMD – please explain the abbreviation and provide CIs and p-value for all SMD; otherwise they are unconfident

Information was added

Line 343: Cabana et al…. – please provide ref No after the author’s name

Done

Line 367: “A 2019 crossover RCT (n=152 subjects (30.1 ± 7.6 years) in adults showed that adding synbiotics to budesonide significantly ameliorated quality of life in persistent allergic rhinitis patients (37) – please provide RR, CI, and a p-value. 

RR and CI are not provide in the paper. We added the p values.

And the next sentence: “The patient population used in this study may be not very representative of allergic rhinitis patients in the overall population (51)” – why, please clarify – unclear.

We added further information to the manuscript : "The patient population used in this study may not be representative for allergic rhinitis patients in the overall population, since symptoms did not respond to usual therapy with antihistamines, antileukotrienes, decongestants and nasal steroids (51)."

Round 2

Reviewer 1 Report

In my opinion the manuscript “Prevention and management with pro-, pre and synbiotics in children with asthma and allergic rhinitis: a narrative review” has been considerably improved and is now suitable for publication. All of my previous comments have been appropriately addressed. I have some other minor suggestions and comments:

- Page 6, line 278-279: check the sentence, I think quality of life should be improved and not decreased .

- Page 6, line 283-284: But -> However

- Page 7, line 337-338: knowlegderhinitis -> knowledge

- Page 8, lime 364-365: Kidilact®:  check the journal policy as regards the trade name of the product

- Page 9, line 407: the full stop between “reported” and “and” should be removed.

- Legend of the tables should include “main” (main studies examining…)

- Table 6: data in the first line is missing

Reviewer 2 Report

The manuscript has been improved as requested.